# Class I Phosphoinositide 3-Kinase *PIK3CA*/p110α and *PIK3CB*/p110β Isoforms in Endometrial Cancer

**DOI:** 10.3390/ijms19123931

**Published:** 2018-12-07

**Authors:** Fatemeh Mazloumi Gavgani, Victoria Smith Arnesen, Rhîan G. Jacobsen, Camilla Krakstad, Erling A. Hoivik, Aurélia E. Lewis

**Affiliations:** 1Department of Biological Science, University of Bergen, 5008 Bergen, Norway; Fatemeh.Mazloumi.Gavgani@uib.no (F.M.G.); Victoria.Arnesen@uib.no (V.S.A.); Rhian.Jacobsen@gmail.com (R.G.J.); 2Centre for Cancer Biomarkers, Department of Clinical Science, University of Bergen, 5021 Bergen, Norway; camilla.krakstad@med.uib.no (C.K.); Erling.Hoivik@uib.no (E.A.H.); 3Department of Gynecology and Obstetrics, Haukeland University Hospital, 5053 Bergen, Norway

**Keywords:** phosphoinositide 3-kinase, *PIK3CA*, *PIK3CB*, p110α, p110β, endometrial cancer

## Abstract

The phosphoinositide 3-kinase (PI3K) signalling pathway is highly dysregulated in cancer, leading to elevated PI3K signalling and altered cellular processes that contribute to tumour development. The pathway is normally orchestrated by class I PI3K enzymes and negatively regulated by the phosphatase and tensin homologue, PTEN. Endometrial carcinomas harbour frequent alterations in components of the pathway, including changes in gene copy number and mutations, in particular in the oncogene *PIK3CA*, the gene encoding the PI3K catalytic subunit p110α, and the tumour suppressor *PTEN*. *PIK3CB*, encoding the other ubiquitously expressed class I isoform p110β, is less frequently altered but the few mutations identified to date are oncogenic. This isoform has received more research interest in recent years, particularly since PTEN-deficient tumours were found to be reliant on p110β activity to sustain transformation. In this review, we describe the current understanding of the common and distinct biochemical properties of the p110α and p110β isoforms, summarise their mutations and highlight how they are targeted in clinical trials in endometrial cancer.

## 1. The Phosphoinositide 3-Kinase Pathway

The phosphoinositide 3-kinase (PI3K) signalling pathway is essential for a myriad of cellular processes and is among the most altered pathways in human cancers, including endometrial cancer [1,2]. Consequently, many cellular processes, coined as hallmarks of cancer [3], are dysregulated due to increased PI3K signalling and contribute to tumour development and progression [4,5,6].

In this review, we aim to give an overview of the mechanism of action of PI3K enzymes in the context of endometrial cancer, in which the PI3K pathway is highly dysregulated. We first summarise current understanding on the two ubiquitously expressed isoforms of class I PI3Ks, p110α and p110β, at the biochemical level and review mutational and other events affecting these two isoforms in endometrial cancer. Finally, clinical trials employing selective inhibitors for p110α and p110β in advanced endometrial carcinomas are summarised and outcomes are reviewed when reported.

### 1.1. Class I PI3K Enzymes

PI3Ks are lipid kinases which phosphorylate the 3’ hydroxyl group on the inositol ring of the glycerophospholipid phosphatidylinositol (PtdIns), or its derivatives, polyphosphoinositides (Figure 1A). This family is divided into three main classes: class I, II and class III depending on their structure and substrate preference [7,8]. Class I PI3Ks, which are further divided into two sub groups, IA and IB, are heterodimers consisting of a catalytic and a regulatory subunit. Class IA consists of a catalytic subunit, p110 (α, β or δ, each encoded by separate genes *PIK3CA*, *PIK3CB* and *PIK3CD*) interacting with one of the regulatory subunits (p85α and splice variants p55α and p50α: encoded by *PIK3R1*, p85β: encoded by *PIK3R2*, p55γ: encoded by *PIK3R3*). The class IB catalytic subunit p110γ (encoded by *PIK3CG*) associates with p84/p87 or p101. In vivo, class I PI3K phosphorylates PtdIns(4,5)*P*_2_ to generate PtdIns(3,4,5)*P*_3_ [9,10] (Figure 1B). Both class II and III generate PtdIns3*P*, whereas class II can also produce PtdIns(3,4)*P*_2_ [11,12,13]. We refer the reader to other reviews on the two latter PI3K classes [14,15] as the focus of this review is on class I PI3Ks.

### 1.2. Activation of Class I PI3Ks and the PI3K Pathway

The activation of class I PI3Ks occurs downstream of the activation of receptor tyrosine kinases (RTK) (for p110α, p110β and p110δ) or G protein coupled receptors (GPCR) (for p110β and p110γ) [7,18] (Figure 2). Upon stimulation of tyrosine kinase receptors, phosphorylation of several tyrosine residues in the intracellular domains leads to the recruitment of the p85/p110 dimer, which releases the inhibition of the p110 catalytic subunit by p85. The mode of activation by GPCR involves the interaction between the Gβ/γ heterodimer and the linker present between the C2 and helical domains in the catalytic unit and their recruitment to the membrane [19,20]. When active and on the membrane, the p110 catalytic subunit can phosphorylate PtdIns(4,5)*P*_2_ to generate PtdIns(3,4,5)*P*_3_ [21]. Target proteins, such as Akt (alias Protein Kinase B), phosphoinositide-dependent protein kinase 1 (PDK1), as well as Sin1 (component of the mammalian target of rapamycin complex 2 (mTORC2), can bind to PtdIns(3,4,5)*P*_3_ through their plextrin homology (PH) domain, thus recruiting them to the plasma membrane [22,23,24,25,26]. Akt is then activated by PDK1 and mTORC2 by phosphorylation at Thr308 and Ser473, respectively [27,28]. Activated Akt, in turn, acts as a downstream signalling molecule, which triggers the activation of multiple other downstream pathways that participate in many cellular processes.

To precisely control the pathway, lipid phosphatases regulate the levels of PtdIns(3,4,5)*P*_3_. The tumour suppressor phosphatase and tensin homolog (PTEN) dephosphorylates PtdIns(3,4,5)*P*_3_ to PtdIns(4,5)*P*_2_, thereby negatively regulating the pathway [29]. Other phosphatases that dephosphorylate PtdIns(3,4,5)*P*_3_ are the 5’ phosphatases, SH2 (Src homology 2)-domain-containing inositol phosphatase (SHIP) 1 and 2, that produce PtdIns(3,4)*P*_2_ [30,31]. The action of these phosphatases allows PtdIns(3,4,5)*P*_3_ to be kept at low levels in resting cells [32].

### 1.3. Properties of PI3K p110α and p110β and Mode of Activation in Cancer

Both p110α and β are ubiquitously expressed, unlike p110δ and p110γ, for which the expression tends to be restricted to the immune system [33]. Due to the context of this review, the focus is on p110α and β. Similarities and differences are summarised for these two isoforms (Table 1 and Figure 3 and Figure A1) and we refer to two reviews for more detailed information [34,35]. Both p110α and β are embryonically lethal in homozygous mice knockouts, suggesting non-redundant functions [36,37]. Following these early studies, homozygous knockin mice with inactivating mutations in the ATP binding site in p110α (D933A) or p110β (D931A) both demonstrated embryonic lethality in an activity-dependent manner, albeit with different penetrance [38,39,40]. This strategy had the advantage of not disrupting the expression and stoichiometry of the catalytic/regulatory PI3K complex. In cell studies, p110α was shown to have a role in cell survival and p110β in DNA synthesis or cell proliferation [41,42,43,44]. Their distinct cellular localisation can be a feature explaining their different cellular roles. p110α is predominantly found in the cytoplasm, but can be detected at very low levels in the nucleus in some cells [45]. p110β is distributed in both the cytoplasm and nucleus including the nucleolus [43,45,46]. p110β has a nuclear localisation signal in the C2 domain, which is absent in p110α ([45], shown in Figure 3 and Figure A1) and plays multiple nuclear roles such as in cell cycle progression, DNA replication, and repair of DNA double-strand breaks [42,43,45,47].

Both enzymes share many structural and biochemical properties, since they use the same substrate to generate the same product, and thus activate the same effector proteins. They are multidomain proteins which have the same domain organisation and share the same mode of activation ([48], Figure 3, see also alignment of p110α and p110β in Figure A1). In particular, class IA p110 catalytic enzymes harbour an adaptor-binding domain (ABD) which interacts with the inter-SH2 linker (iSH2) of p85 and promotes stability. In addition, interaction of the N-terminal SH2 domain and iSH2 in p85 with the C2 and helical domain of p110 blocks basal catalytic activity [48]. Phosphorylated tyrosine residues on activated RTKs bind to the SH2 domains of p85s and release the inhibitory interaction of these domains with the p110 catalytic subunit. In p110β, an additional inhibitory contact between the C-terminal catalytic domain of p110 and the C-terminal SH2 domain of p85, entails a different mechanism to release the inhibitory interaction by phosphorylated RTKs [48,49].

In cancer, alterations in both catalytic isoforms have been reported, albeit at different frequencies [6]. Since the observation of high mutational frequency in human cancers suggests *PIK3CA* as a driver, much research effort has focused on this gene [50]. The most common activating mutations in *PIK3CA* are found in the helical domain (E542 and E545) and the kinase domain (H1047) and activate p110α through different mechanisms, such as reducing the inhibitory effect of p85 or facilitating the interaction with the lipid membrane [51,52,53]. In addition, mutations in p85 can lead to the stimulation of the p110 subunit, as shown not only for p110α but also for p110β [54,55]. A recent study by Thorpe et al. demonstrated that a decrease in p85α resulted in elevated p85-p110 complex signalling in vitro, correlating to increased tumour development in breast cancer mouse models [56]. As for *PIK3CB*, mutations are in general less frequent compared to *PIK3CA*, and *PIK3CB* can be amplified or overexpressed in some solid tumours [6]. Importantly, p110β can promote oncogenic transformation when overexpressed in its wild type (WT) form, in contrast to p110α, which requires the presence of activating mutations [57].

A few key biochemical differences help explain the distinct mode of contribution in tumourigenesis for both isoforms. Firstly, a critical difference was identified in the C2 domain and this may explain their differential activation mode. Indeed, N345 in p110α is involved in hydrogen bond interactions with the iSH2 domain of p85. This residue aligns with K342 in p110β, which corresponds to oncogenic mutations found in p110α in some cancers, from N345 to K345 ([58], shown in Figure 3 and Figure A1). This one residue difference decreases the inhibitory interaction of p110β with p85 [58] and may explain the transforming capacity, at least partially, of WT p110β compared to p110α, which requires mutation for transformation [57]. Secondly, in contrast to p110α, p110β can be activated by GPCRs through its direct association with the G protein subunits β/γ [19,59,60,61,62] and through the RAC (ras-related C3 botulinum toxin substrate) small GTPAse [63]. Importantly, activation of p110β by GPCR was required for cell invasiveness in breast cancer cells [62]. Cell proliferation was also dependent upon GPCR-mediated activation of p110β in PTEN-negative prostate and endometrial cancer cells, but not PTEN-positive cells [19].

A few mutations have recently been validated as oncogenic for *PIK3CB*. These include the helical domain mutation (E633K), identified in a Her2-positive breast tumour, which leads to increased p110β association with the membrane and increased basal activity [64]. A mutation in the *PIK3CB* kinase domain (D1067V), was shown to occur in several tumour types at low rates [65]. A gain-of-function mutation in the kinase domain of *PIK3CB* (E1051) was found to be an oncogenic driver and to promote cell growth and migration [66]. While studying the selective p110β inhibitor (GSK2636771) on solid tumours, a patient with castrate-resistant prostate cancer was found to have a mutation (L1049R) in the *PIK3CB* gene [67]. Functional characterisation of variants identified from cancer genome sequencing showed malignant transformation due to a rare mutation (A1048V) in the *PIK3CB* gene [68]. p110β also plays a role in the resistance of tumours to inhibitors of p110α [69,70]. A mutation (D1067Y) in *PIK3CB* has also been detected in cells resistant to pan-PI3K inhibition, which induced the activation of the PI3K signalling pathway [71]. Furthermore, p110β activity contributes to tumour progression and its expression correlates with poor prognosis and metastasis in breast cancer [59,72].

In 2008, a study which aimed to find the lipid kinase required to sustain and drive PI3K signalling in PTEN-deficient cancers showed that p110β played an essential role in these cancers and that its lipid kinase activity was required [73]. Later studies showed that, indeed, p110β activity is required in PTEN-deficient breast, prostate and haematological tumour growth [61,74,75], but not in other tissues which reported reliance on p110α or both isoforms [76,77]. A recent study showed that the PI3K pathway in PTEN-null tumours that rely on p110β are regulated by the Crk-like protein (CRKL) adaptor protein [78]. The study showed that CRKL has a preference to associate with p110β over p110α, and through its interaction with p110β/p85, it regulates the PI3K signalling pathway.

## 2. Alteration of the PI3K Pathway in Endometrial Cancer

### 2.1. Endometrial Cancer

Endometrial cancer arises from lesions in the lining of the uterus (also known as the uterine corpus) which are, in up to 95% of cases, carcinomas with the remaining being sarcomas [82,83]. This gynaecological cancer is highly prevalent in developed countries, is highly associated with obesity and its incidence is rising [84,85]. Endometrial carcinomas have different histologies and were traditionally divided into two subtypes, type I and type II, according to the Bokhman classification [86]. Type I accounts for the majority of endometrial cancers and consists of low-grade tumours of endometrioid histology which are positive for hormone receptors and have a good prognosis [86,87]. These tumours are often referred to as endometrioid endometrial cancers (EEC). In contrast, type II tumours, also known as non-endometrioid (NEEC), are less common, of high grade, and hormone receptor negative with a poor prognosis [86,87]. Type II can display the following histologies: serous adenocarcinomas, clear cell adenocarcinomas and carcinosarcomas [86,87]. More recently, large-scale sequencing studies of primary endometrioid tumours (UCEC), initiated by The Cancer Genome Atlas (TCGA) genome network, suggested a classification of endometrial cancers into four major groups depending on different molecular signatures: (1) Polymerase ε (POLE), ultramutated with the highest survival outcome; (2) microsatellite instability (MSI), hypermutated with an intermediate outcome; (3) low copy number (endometrioid) with an intermediate outcome; and (4), high copy number (serous-like) with the poorest outcome [88]. This molecular classification has been further compared to and integrated with previous classification properties such as grade, genetic alterations and histology [87]. Type I tumours are hence considered endometrioid tumours with the following molecular characterisation: ultramutated POLE, microsatellite instability (MSI), hypermutated and a low copy number. Type II, including serous or clear cells, consists of the high copy number group. Specific molecular alterations representing each group can now be selected to evaluate endometrial tumours for diagnosis.

### 2.2. Alteration of PI3K PIK3CA and PIK3CB in Endometrial Cancer

The PI3K pathway is the most frequently altered in endometrial cancer [87,88,89]. The most frequent mutations are found in the *PTEN*, *PIK3CA* and *PIK3R1* genes, particularly in type I endometrioid tumours [88,90,91,92,93]. The distribution of mutations of these genes obtained from the endometrial cancer TCGA study is shown in Figure 4A [88,94,95]. PTEN loss is the most frequent alteration in endometrioid tumours with a frequency of up to 80% and mutations are often seen in hyperplasias, considered precursor lesions to endometrial cancer [96,97,98,99,100]. *PIK3CA* is the second most mutated gene, with overall frequencies of 25% of tumours according to the Catalog of Somatic Mutations In Cancer (COSMIC, v86 [101]) as well as of 51% and 53% in TCGA-UCEC (release 13) and the TCGA endometrial cancer genomic data, respectively ([88,95] Figure 4A,B and Figure A1). Mutations occur in both type I endometrioid and serous carcinomas ([88,92,98,102,103,104,105], Figure 4A). The effect of *PIK3CA* mutations on clinical variables is conflicting, as positive and negative associations have been shown in relation to survival [106]. However, in a recent study of *PIK3CA* mutations in exon 9, corresponding to part of the helical domain, mutations were associated with poor survival [106]. *PIK3CA* mutations occur early in endometrial cancer progression and are of high clonality from primary lesions to metastasis [99,100,106]. Other types of alterations in *PIK3CA* have also been reported. *PIK3CA* mRNA levels were higher in non-endometrioid tumours and increased from primary tumours to their corresponding metastasis, and high mRNA levels correlated with lower survival [44,106]. In addition, *PIK3CA* amplification, which is associated with high level PI3K signalling, correlates with NEEC/type II aggressive endometrial cancer phenotype [88,104,107,108] (Figure 4A).

The frequency of *PIK3CB* mutation is lower compared to the *PIK3CA* gene, with reported frequencies of 2% (COSMIC), 10% (TCGA-UCEC, release 13) and 8% (TCGA endometrial cancer genomic data [88,95], Figure 4B). So far, only two studies have reported the occurrence of *PIK3CB* mutations in endometrial cancer, D1067V and A1048V within the kinase domain, and shown that they are oncogenic [65,68]. Other mutations were detected in the TCGA genomic study, highlighting other potential oncogenic mutations, which are to date uncharacterised for their oncogenic properties (Figure 4C). A more common alteration is an increase of its mRNA levels reported in two studies [44,109] and by COSMIC (v86) with a frequency of 6.8%. In particular, the mRNA levels of *PIK3CB* were shown to be higher in grade 1 endometrioid endometrial lesions when compared to complex hyperplasias and remained high in higher grades as well as in NEEC tumours [44]. Importantly, high levels of *PIK3CB* mRNA correlated with lower survival [44]. In cell line studies, the protein levels of p110β, but not of p110α, were elevated in endometrial cancer cells compared to non-transformed cells [44]. In addition, gene amplification was also detected, particularly in serous tumours (Figure 4A). Considering the transforming ability of p110β in its WT state, overexpression of this isoform may account for tumour development in endometrial cancer. Consistently, knock down of p110β in endometrial cancer cell lines induced cell death [109].

## 3. Targeting p110α and p110β in Endometrial Cancer

About 20% of endometrial tumours recur and respond poorly to currently available systemic therapy. Because the PI3K pathway is frequently altered in endometrial cancer, it is an attractive target for therapy, as recently validated by large-scale genomic sequencing reports [88,89,100,110,111]. PI3K chemical inhibitors have hence been tested in pre-clinical studies and used to target tumours with aberrant activation of the pathway.

### 3.1. PI3K Inhibitor Studies in Endometrial Cancer Cell Lines

A number of pre-clinical studies have been performed using endometrial cancer cell lines to test the efficacy of pan-PI3K inhibitors as well as selective p110α and p110β inhibitors [44,112,113]. *PIK3CA* mutant cancer cells were more prone to respond to pan-PI3K or selective p110α inhibition compared to WT cells [44,113].

Considering the importance of *PIK3CB*/p110β in PTEN-deficient tumours and the high frequency of PTEN mutations in endometrial cancer, the effect of p110β selective inhibition was evaluated in endometrial cancer cell lines, with or without PTEN loss [44,113]. Selective inhibition of p110α and p110β led to different effects on cell signalling and cell outcomes. p110α activity was correlated with cell survival and its inhibition led to decreased cell survival in *PIK3CA* mutant cells but not in WT cells [44,113]. In contrast, p110β inhibition had no effect on cell survival but rather decreased cell proliferation in PTEN-deficient cells [44,113]. Considering that endometrial tumours can harbour alterations in both *PIK3CA* and *PTEN*, combination treatment with both p110α and p110β may increase response efficacy compared to monotherapy [113]. The determination of the genetic status of *PIK3CA* and *PTEN* is of great importance for the most appropriate personalised treatment. In addition, the presence of genetic alteration in *PIK3CB* as well as increased levels of p110β may also influence treatment outcome in a few cases.

### 3.2. Clinical Trials in Endometrial Cancer

The pan-PI3K inhibitor Pictilisib (BKM120) was administered to patients with advanced or metastatic endometrial cancer but adverse side effects were observed, and the clinical trial was stopped ([114] NCT01397877). In contrast, Pilaralisib (SAR245408; XL147), another pan-PI3K inhibitor, used in a clinical trial of advanced or recurrent endometrial carcinomas, did not show any severe adverse events on patients but had only minimal anti-tumour activity ([115] NCT01013324). Finally, a trial planning to use the pan-PI3K inhibitor Copanlisib (BAY 80-6946) in patients with persistent or recurrent endometrial cancer with *PIK3CA* hotspot mutations was suspended (NCT02728258). Selective inhibitors for p110α and p110β are being used in clinical trials but the results in patients with endometrial cancer are scarce as they are part of large studies including different types of cancer. Patients with *PIK3CA*-altered advanced cancer were treated with the p110α selective inhibitor Alpelisib (BYL719) and one patient with endometrial cancer showed a complete response and another showed a partial response to treatment ([116], NCT01219699). However, the study does not clearly state the total number of patients with this type of cancer and the respective mutational status.

A phase I trial with the p110β selective inhibitor GSK2636771 in PTEN-deficient solid advanced cancers, including three patients with endometrial cancer, was completed and reported (Reference [67] and Table 2). One patient out of three with endometrial cancer benefitted from progression-free disease for 33 weeks. Intriguingly, this patient did not harbour any mutation or gene copy variation in *PTEN*, *PIK3CA*, *PIK3CB* or *AKT2*. As we are writing this review, other clinical trials are currently recruiting patients with advanced solid tumours harbouring PTEN loss or with *PIK3CB* mutation and/or amplification and may include patients with endometrial cancer (Table 2).

## 4. Conclusions

Alteration in the PI3K pathway is undoubtedly a key event in endometrial carcinomas with differences in molecular genetic features throughout histologies, stages and grades. Much of the research and clinical efforts have focused on *PIK3CA*/p110α but are starting to include *PIK3CB*/p110β in light of its association with loss of PTEN, the most frequent genetic alteration in endometrial carcinomas. To date, clinical trials include few patients with endometrial cancer, making it challenging to draw any reliable conclusions on the correlation between the genetic alteration status of the PI3K pathway and outcomes. Still, a positive outcome was reported for one patient treated with a selective p110β inhibitor who responded well to the treatment [67]. Several clinical trials with different p110β inhibitors are recruiting patients with advanced solid cancers with PTEN loss but also harbouring *PIK3CB* mutations or amplification. However, it is not yet known whether these will include endometrial carcinomas.

## Figures and Tables

**Figure 1 ijms-19-03931-f001:**
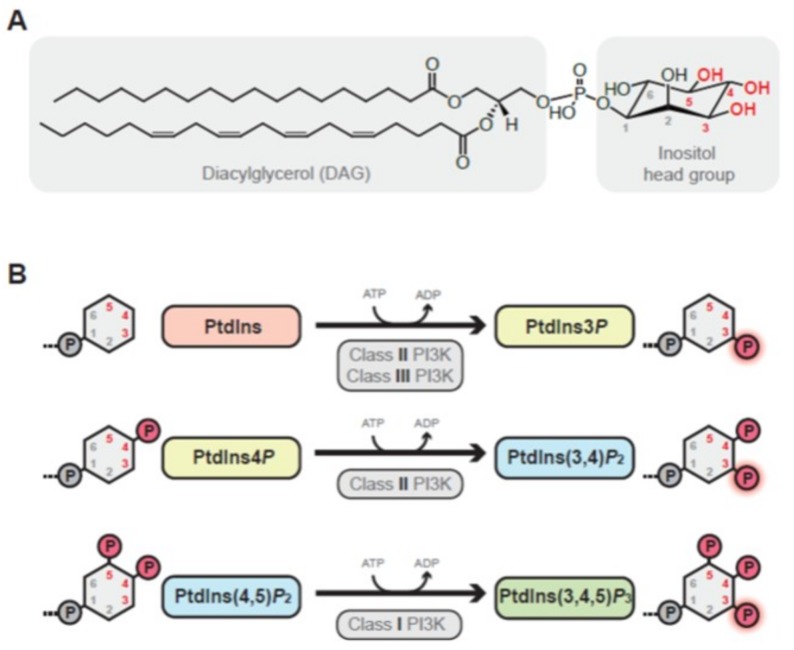
Chemical structure of phosphatidylinositol and phosphoinositide 3-kinase (PI3K) enzyme reactions. (**A**) Phosphatidylinositol chemical structure PI(18:0/20:4(5Z,8Z,11Z,14Z)) downloaded from the LIPID MAPS Structure Database (LM ID: LMGP06010010) [16,17]. Hydroxyl groups located at positions 3, 4 and 5 of the myo-inositol head group that are targeted by phosphorylation by polyphosphophoinositide kinases are highlighted in red. (**B**) Main enzymatic reactions carried out by the different phosphoinositide 3-kinase (PI3K) classes. Only the inositol head groups are shown with sites of phosphorylation marked in red. Phosphate groups are indicated with a circled P.

**Figure 2 ijms-19-03931-f002:**
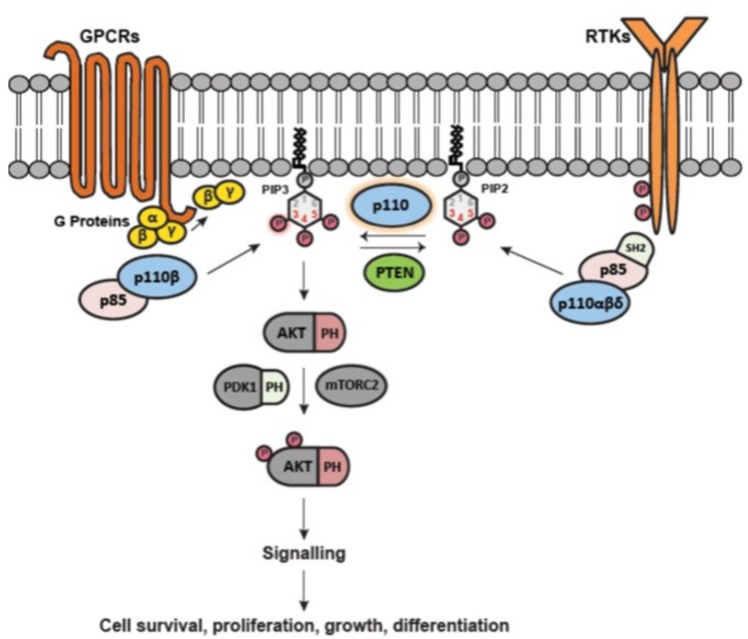
Class IA PI3K activation. Class IA PI3Ks consist of a catalytic subunit (p110) and a regulatory subunit (p85). Upon stimulation of receptor tyrosine kinase (RTK) or G-protein-coupled receptor (GPCR), the p85-p110 complex is targeted to the membrane via different mechanisms (due to tyrosine phosphorylation of RTKs or interaction with G-proteins β/γ). The p85 subunit loses its inhibitory effect on the catalytic activity of p110, and thereafter the p110 subunit phosphorylates phosphatidylinositol (4,5)-bisphosphate (PIP2) to generate phosphatidylinositol (3,4,5)-triphosphate (PIP3). PIP3 then targets proteins containing the pleckstrin homology (PH) domain—such as AKT, phosphoinositide-dependent kinase 1 (PDK1), as well as Sin1 (not shown), part of the mammalian target of rapamycin complex 2 (mTORC2) and locates them to the plasma membrane. AKT is then phosphorylated on Thr308 and Ser473 by PDK1 and mTORC2 respectively. Once these proteins are activated at the membrane, they trigger a signalling cascade that leads to multiple cellular functions. Phosphate groups are indicated with a circled P.

**Figure 3 ijms-19-03931-f003:**
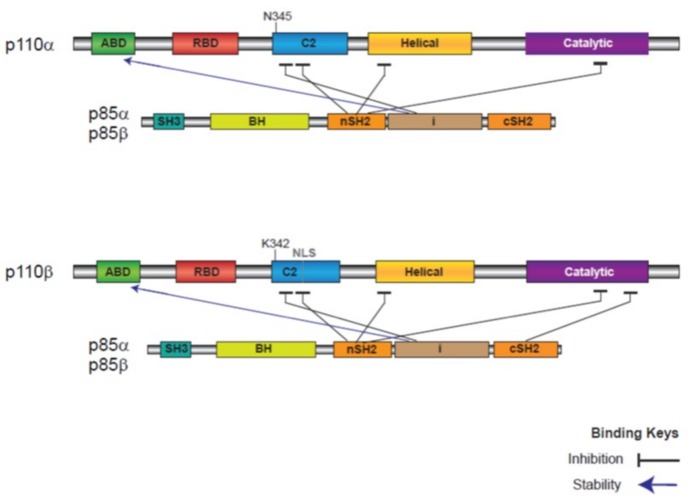
Domain structure of class IA PI3K catalytic and regulatory subunits. Domain structure and interaction of the catalytic subunits p110α and p110β and regulatory subunits p85α and p85β. The regulatory p85 subunit binds to p110 to inhibit and stabilise the lipid kinase. The interlinker domain of p85 binds to the ABD domain of the p110 subunit to stabilise the kinase (shown with an arrow). On the other hand, the nSH2 and interlinker domains in p85 have an inhibitory effect on both p110α and β. In addition, the p110β isoform can be inhibited by the cSH2 domain of p85. The figure is adapted from Reference [18]. Abbreviations: ABD, adaptor binding domain; RBD, Ras-binding domain, C2; protein-kinase-C-homology-2 domain, Helical; helical domain, Catalytic; kinase domain; NLS, nuclear localisation sequence; SH, Src homology; BH, breakpoint-cluster region homology; nSH2, N-terminal SH2; cSH2, C-terminal SH2, i, interlinker SH2 coiled-coiled domain.

**Figure 4 ijms-19-03931-f004:**
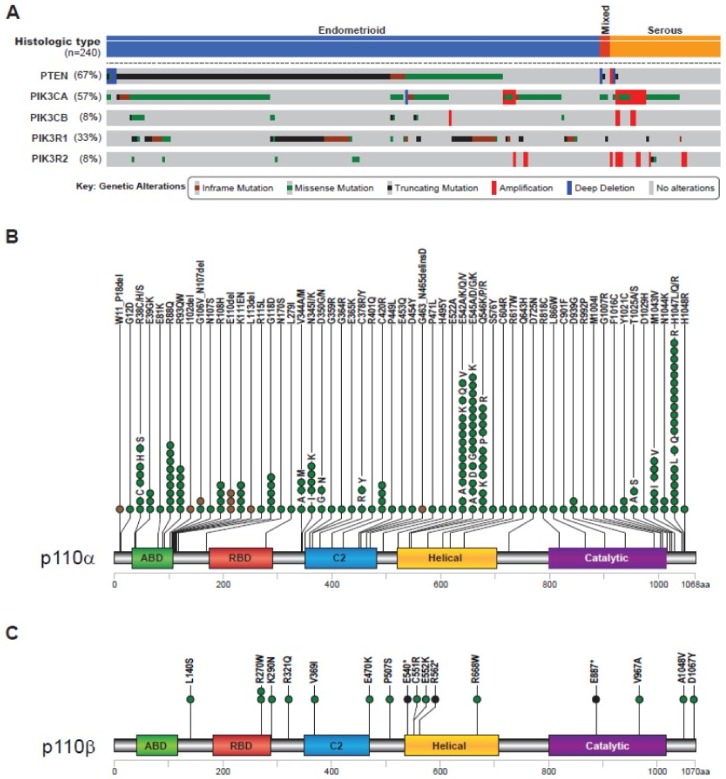
Overview of genomic alterations in PI3K pathway genes in endometrial cancer. (**A**) Overview of alterations in selected PI3K pathway genes in endometrial cancer, showing mutations and copy-number alterations, as well as the frequency of alterations. (**B**) Overview of mutations in *PIK3CA* displayed on the p110α protein structure. Note the hotspot mutational sites in p.E524, p.E545, p.Q546 and p.H1047, i.e., those with the highest number of lesions and the corresponding mutation. (**C**) Overview of mutations in *PIK3CB* displayed on the p110β protein structure by amino acid position. Panels were generated using the cBioPortal [94,95] with modifications with data from The Cancer Genome Atlas (TCGA) from a total of 240 patients with endometrial cancer [88]. Each dot corresponds to the occurrence of a specific mutation in a lesion. Mutations in (**B**) and (**C**) are colour-coded according to the key shown in (**A**) (green for missense mutations and brown for other in-frame mutations such as insertion and deletion). Abbreviations: ABD, adaptor binding domain; RBD, Ras-binding domain, C2; protein-kinase-C-homology-2 domain, Helical; helical domain, Catalytic; kinase domain.

**Table 1 ijms-19-03931-t001:** Differences and similarities between p110α and p110β.

Property Description	p110α	p110β
Gene name	*PIK3CA*	*PIK3CB*
Regulatory subunit ^1^	p85α, p55α, p50α	p85α, p55α, p50α
	p85β	p85β
	p55γ	p55γ
Cellular localisation	cytoplasm	cytoplasm, nucleoplasm, nucleolus
Receptor activation	RTKs	GPCRs (dominant) and RTKs
Mutations in carcinomas	frequent	rare

^1^ p110α and p110β can, in theory, form heterodimers with any of the regulatory subunits, depending on the tissue of interest [33,79,80]. p85α and p85β are ubiquitously expressed, whereas the expression of the shorter isoforms p55α, p50α or p55γ is restricted to certain tissues [33,81]. Abbreviations: RTK: receptor tyrosine kinase. GPCR: G protein-coupled receptor.

**Table 2 ijms-19-03931-t002:** Completed and planned clinical trials using p110β inhibitors.

Drug Name	Molecular Condition for Trial	Types of Cancer	Phase	ID Number
GSK2636771	PTEN deficiency	advanced solid tumours	I	NCT01458067 (completed and results published [67])
GSK2636771	PTEN loss, mutation or deletion	Advanced-stage refractory solid cancers	II	NCT02465060 (recruiting patients)
AZD8186	PTEN loss, mutation or deletion	Advanced-stage refractory solid cancers	II	NCT02465060 (recruiting patients)
AZD8186 (with docetaxel)	PTEN loss or mutation, or *PIK3CB* Mutation	Advanced-stage solid cancers metastatic or unresectable	I	NCT03218826 (recruiting patients)

Information on clinical trials was retrieved from www.clinicaltrials.gov.

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
