# Peer review of "Class I Phosphoinositide 3-Kinase PIK3CA/p110α and PIK3CB/p110β Isoforms in Endometrial Cancer"

_ijms, 2018, doi:10.3390/ijms19123931_

Round 1

Reviewer 1 Report

In this review, the authors describe the potential role of ubiquitous class I PI3K in endometrial cancers.

This review is nicely written and have high relevance both in basic and clinical research.

Here are my minor comments:

- l 95: cite the abbreviation;

- figure 2: describe mTORC2 as a complex with proteins with PH domains (Sin1)  - ref : Liu P, cancer discovery 2015.

- l 104 also cite knock - in mice (Graupera, et al, Nature 2008; Guillermet-Guibert J, Plos genetics

2015) ; the embryonic lethality is different.

- l 108 in Kumar A, Mol cell biol 2011, p110a is expressed in the nucleus albeit at low levels and not in all cells - please discuss.

- l153-154: could this information be inserted in figure 3 - if read by a non specialist, the scheme can be interpreted as p85 has more inhibitory effect on p110beta.

- table 1 - provide reference to p110a/b-p85s differential coupling (Geering B, PNAS 2007)

- figure 4 - the legend is too minimal, it is difficult to understand why it is a hotspot - to what corresponds the dots?

- appendix A- the color coding is difficult to read.

Author Response

Response to Reviewer 1 Comments

We thank the reviewer for many helpful comments and suggestions. We feel that the revised manuscript has greatly benefited from the reviewer’s input. Here are our detailed answers to each point raised. Corresponding changes are shown highlighted in yellow in the revised manuscript. Additional changes are shown with track changes. The whole manuscript has also been thoroughly proof-read by professionals.

Point 1 - l 95: cite the abbreviation;

Response 1: we have added the abbreviation, now on line 114.

Point 2- figure 2: describe mTORC2 as a complex with proteins with PH domains (Sin1)  - ref : Liu P, cancer discovery 2015.

Response 2: Please find the added citation and relevant text on lines 90-1 and in the figure 2 legend.

Point 3 - l 104 also cite knock - in mice (Graupera, et al, Nature 2008; Guillermet-Guibert J, Plos genetics 2015) ; the embryonic lethality is different.

Response 3: Please find the added citations and relevant text on lines 132-6.

Point 4 - l 108 in Kumar A, Mol cell biol 2011, p110a is expressed in the nucleus albeit at low levels and not in all cells - please discuss.

Response 4: We have changed this sentence, now located on lines 138-40 to: “p110α is predominantly found in the cytoplasm, but can be detected at very low levels in the nucleus in some cells [45]. p110β is distributed in both the cytoplasm and nucleus including the nucleolus”

Point 5 - l153-154: could this information be inserted in figure 3 - if read by a non specialist, the scheme can be interpreted as p85 has more inhibitory effect on p110beta.

Response 5: We have added the N345 and K342 in p110a and p110b respectively in Figure 3

Point 6 - table 1 - provide reference to p110a/b-p85s differential coupling (Geering B, PNAS 2007)

Response 6: This information as well as the references are now found in table 1

Point 7- figure 4 - the legend is too minimal, it is difficult to understand why it is a hotspot - to what corresponds the dots?

Response 7: The legend has been clarified.

Point 8 - appendix A- the color coding is difficult to read.

Response 8: This has been corrected.

Reviewer 2 Report

I was pleased to review the Manuscript titled “Class I phosphoinositide 3-kinase PIK3CA/p110α and PIK3CB/p110β isoforms in endometrial cancer” (ijms-389654). The topic of this paper is interesting, so it may be of great interest to the readers of International Journal of Molecular Sciences. However, the Manuscript should be further improved.

 Proof-reading by native English speaker is mandatory, in order to improve readability and correct several typos.

Author Response

The whole manuscript has also been thoroughly proof-read by professionals. Changes are shown with track changes in the revised manuscript. Additional changes requested by reviewer 1 are highlighted in yellow.
